# A Compact Ultrafast Electron Diffractometer with Relativistic Femtosecond Electron Pulses

**Jinfeng Yang [1,*]**, **Kazuki Gen [1]**, **Nobuyasu Naruse [2]**, **Shouichi Sakakihara [1]** and **Yoichi Yoshida [1]**

[1] The Institute of Scientific and Industrial Research, Osaka University, Osaka 567-0047, Japan; gen-kazuki81@sanken.osaka-u.ac.jp (K.G.); shouichi@sanken.osaka-u.ac.jp (S.S.); yoshida@sanken.osaka-u.ac.jp (Y.Y.)

[2] School of Medicine, Shiga University of Medical Science, Shiga 520-2192, Japan; naruse@belle.shiga-med.ac.jp

* Correspondence: yang@sanken.osaka-u.ac.jp; Tel.: +81-6-6879-4285

**Abstract:** We have developed a compact relativistic femtosecond electron diffractometer with a radio-frequency photocathode electron gun and an electron lens system. The electron gun generated 2.5-MeV-energy electron pulses with a duration of $55 \pm 5$ fs containing $6.3 \times 10^4$ electrons per pulse. Using these pulses, we successfully detected high-contrast electron diffraction images of single crystalline, polycrystalline, and amorphous materials. An excellent spatial resolution of diffraction images was obtained as $0.027 \pm 0.001$ Å$^{-1}$. In the time-resolved electron diffraction measurement, a laser-excited ultrafast electronically driven phase transition in single-crystalline silicon was observed with a temporal resolution of 100 fs. The results demonstrate the advantages of the compact relativistic femtosecond electron diffractometer, including access to high-order Bragg reflections, single shot imaging with the relativistic femtosecond electron pulse, and the feasibility of time-resolved electron diffraction to study ultrafast structural dynamics.

**Keywords:** ultrafast electron diffraction; relativistic electron pulse; femtosecond; single shot imaging; time-resolved; single crystals; polycrystals; amorphous; structural dynamics

## 1. Introduction

Ultrafast electron diffraction (UED) with femtosecond temporal resolution is a powerful tool to observe directly dynamic processes in materials science, chemistry, and biology. Most widely used compact UEDs have been developed with a photocathode dc-acceleration-based electron gun to generate nonrelativistic pulses with energies of less than 100 keV [1–5]. In the nonrelativistic UEDs, space-charge effect due to the Coulomb repulsion of electrons in pulse is a serious problem. The space charge effect broadens the pulse duration and increases the energy spread during the propagation between the photocathode and sample. There are two approaches to circumvent this problem: One is the use of a low number of electrons per pulse, i.e., a 100-fs long pulse containing only two electrons [6]. This approach is limited to the study of reversible processes only. Another approach is the use of a short distance between the photocathode and sample to minimize the effects of space charge. In this approach, the 600-fs pulses with 6000 electrons per pulse were generated [7]. However, the pulse duration of electrons is long.

In recent years, relativistic-energy UEDs were developed with an advanced particle accelerator technology of photocathode radio-frequency electron gun (rf gun) [8–15]. Recently, an ultrafast electron microscope (UEM) using the rf gun has been developed at Osaka University [16,17]. The rf gun was operated with a high peak rf electric field of 100 MV/m on the photocathode, which is ten times higher

than that of the dc gun. The electrons emitted from the photocathode are then quickly accelerated into the relativistic energy region to reduce the effect of space charge, yielding ultrashort pulses with a large number of electrons per pulse, i.e., a 100 fs long pulse containing $10^6 \sim 10^7$ electrons at an energy of 3 MeV [18,19]. Moreover, the high-energy electrons significantly increase the extinction distance of elastic scattering [20]. Our previous UED study indicates that the kinematic theory with the assumption of single elastic scattering events can be applied in the relativistic UED [21,22]. This enables one to easily explain structural dynamics from the experimental results.

However, in the current relativistic UED instruments, a relatively long distance of 4 m or more between the sample and detector is required to record the acceptable diffraction images [13–15]. Many significant efforts, including the generation of high-quality electron pulses and the construction of compact UED with high tempo-spatial resolution and good signal-to-noise ratio, are required for this approach to be realized. Here, we report a compact relativistic UED instrument using an rf gun and an electron lens system. The lens system is used to form the electron diffraction image and projects it on the detector, yielding a short distance between the sample and detector. We present the demonstrations on the electron diffractions of single crystalline, polycrystalline, and amorphous materials, and the time-resolved UED measurement of single crystalline silicon using relativistic femtosecond electron pulses.

## 2. Relativistic UED Instrument

The compact relativistic UED was developed with a 1.6-cell 2856-MHz-rf photocathode electron gun and an electron lens system, as shown in Figure 1. The rf gun has been designed and fabricated with new considerations and improvements [23–26], including the use of circular-shape rf cavities and a large-size iris of 25 mm in diameter between the half-cell and full-cell to minimize the rf-induced emittance and energy spread, gold-brazing the photocathode plate with the half-cell to reduce dark current emission, and the increase of water cooling channels to stabilize the cavity temperature. A copper photocathode was used and illuminated by the third harmonic of a Ti:Sapphire laser (wavelength of 266 nm, pulse duration of 54 fs in root mean square (RMS)). The diameter of the laser on the cathode was 0.3 mm focused by an optical lens. The rf cavities were driven by a klystron with a peak power of 4 MW. The repetition rate of the electron pulses was 10 Hz.

After the rf gun, a solenoid lens and a condenser lens were used to parallel the electrons onto the sample. A condenser aperture made of molybdenum metal with four pinholes was installed between the solenoid and condenser lenses to stop divergent electrons. The diameters of pinholes were 0.3, 0.5, 1, and 2 mm. The convergence angle of electrons with the 0.3-mm diameter pinhole was 55 μrad, which is effectively a parallel electron beam. Moreover, the convergent-beam electron diffraction (CBED) measurement can be utilized by focusing the electron pulses onto the sample with the condenser lens.

The sample was placed downstream of the condenser lens and adjusted precisely by a 5-axis motorized stage. In the experiments, a parallel electron beam was used to illuminate the sample. After the sample, a diffraction lens and a projector lens were used for electron diffraction imaging. We adjusted the projector lens, so that the back-focal plane of the diffraction lens acted as the object plane of the projector lens. The diffraction images were then projected onto a viewing screen (scintillator). A Tl-doped CsI scintillator equipped with a fiber optic plate (Hamamatsu Photonics) and a lens-coupled electron-multiplying charge-coupled device (EMCCD) camera with $1024 \times 1024$ pixels were applied for the electron diffraction measurement. The size of the scintillator was $50 \times 50$ mm$^2$. This scintillator has been tested and successfully used to detect relativistic electron diffraction images with a spatial resolution of 50 μm in our previous UED study [11]. The distance between the sample and scintillator was 1.6 m. The sample can be cleaning in a separation vacuum chamber (sample preparation room) before the measurement.

In the time-resolved UED experiments, the samples were excited by femtosecond laser photons, as shown in Figure 1. The Ti:Sapphire laser oscillator was time-synchronized to an external rf signal of 79.3 MHz, which is produced by dividing the accelerating 2856-MHz rf signal by 1/36. The time jitter

between the laser pulse and the 79.3-MHz rf phase was 61 fs RMS [16], being approximately equal to the time jitter between the pump laser pulse and the electron pulse.

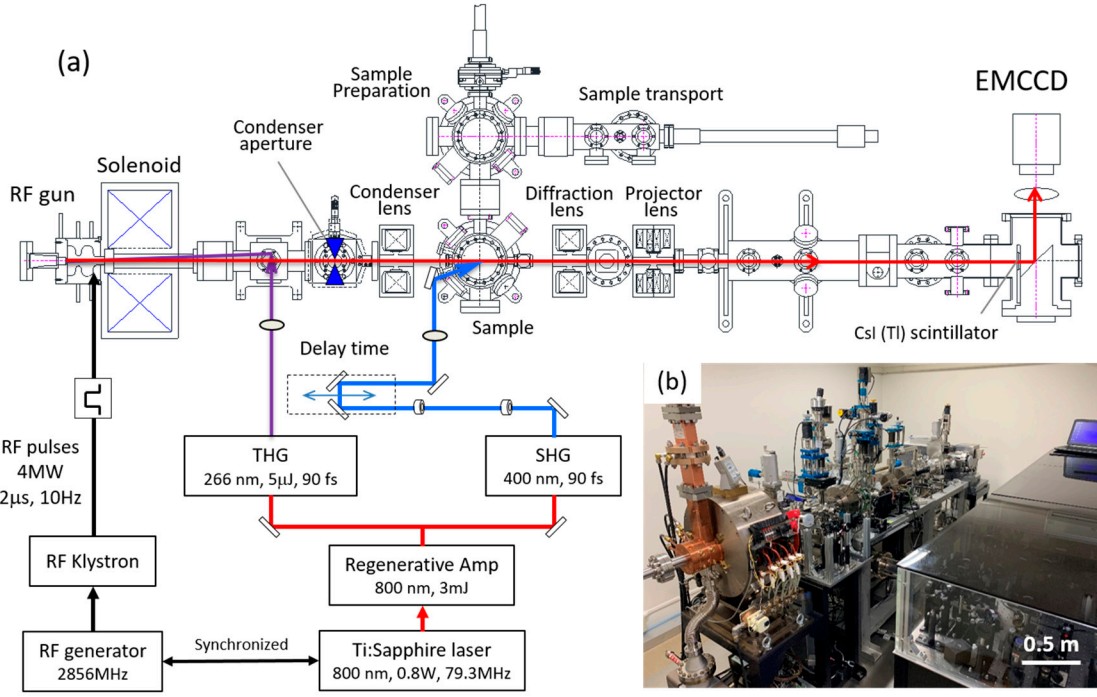

**Figure 1.** (**a**) Schematic and (**b**) photo of a compact relativistic ultrafast electron diffraction (UED) instrument using a photocathode rf gun, where third-harmonic generator (THG) is used to generate third harmonic of the Ti:Sapphire laser for generating electron pulses and second-harmonic generator (SHG) is used to generate second harmonic of the Ti:Sapphire laser for exciting the sample.

The laser pulses were amplified by a regenerative amplifier (Spitfire Ace, Spectra-Physics, Osaka, Japan). The maximum laser energy was 3 mJ. The repetition rate was 1 kHz. The output of the amplifier was divided into two beams: One was converted to 266 nm laser with third-harmonic generator (THG) for the electron pulse generation. Another was used as a pump laser to excite the sample. The 400-nm pump laser of the output of second-harmonic generator (SHG) was used in the time-resolved measurement as described in Section 3.4. Two optical choppers installed in the pump laser beam line were used to reduce the repetition rate of pump pulses from 1 kHz to 10 Hz. The laser spot size on the sample was 0.8 mm focused by an optical lens. The incident maximum fluence was >100 mJ/cm$^2$. The pulse duration of the pump laser was 54 fs RMS. The incident angle of the pump laser light was 14 degrees from the electron beam line. All UED components and the Ti:Sapphire laser are localized on a vibration-controlled board. The size was $3 \times 3$ m$^2$, which is effectively a compact relativistic-energy UED instrument. Although relativistic-energy electron diffraction imaging can be realized in the relativistic UEM [19], the presented relativistic UED instrument exhibits many excellent advantages, including ultrahigh vacuum at $10^{-9}$ Pa, sample preparation, and a large sample room for various in-situ observations with heating, pressuring, electric field, and liquid or gas flow.

## 3. Results and Discussion

### 3.1. Single Shot Electron Diffraction of Single Crystalline Materials

In the single shot electron diffraction imaging with the relativistic femtosecond electron pulse, both the samples of a (100)-orientated single crystalline gold film with a thickness of 10 nm (TAAB Laboratories Equipment Ltd., Aldermaston, UK) and a (001)-orientated single crystalline silicon (Si) film with a thickness of 35 nm were used. The single crystalline Si sample was produced from a

60-µm-thick Si wafer by photolithography and plasma etching. The electron pulses were collimated by the condenser aperture with a diameter of 0.3 mm. The number of electrons per pulse was $6.3 \times 10^4$ (10 fC) on the samples. Figure 2 shows the single shot electron diffraction images of the single-crystalline gold and Si samples observed by a single electron pulse. The electron energy was 2.5 MeV. The pulse duration was not measured in the experiments; however, we estimated it to be 55 ± 5 fs RMS for $6.3 \times 10^4$ electrons per pulse at 2.5 MeV by the theoretical simulation using a general particle tracer (GPT) code [27] at the rf gun launch phase of 30° with the 266-nm and 54-fs laser illumination. The calculated duration of electron pulses was approximately equal to the laser pulse duration, indicating the effects of space charge are negligible in the relativistic electron pulses. The lines in Figure 2c,d represent the intensity profiles of Bragg peaks along the $0\bar{2}0$ and 020 diffraction spots in the gold sample and along the $2\bar{2}0$ and $\bar{2}20$ diffraction spots in the Si sample.

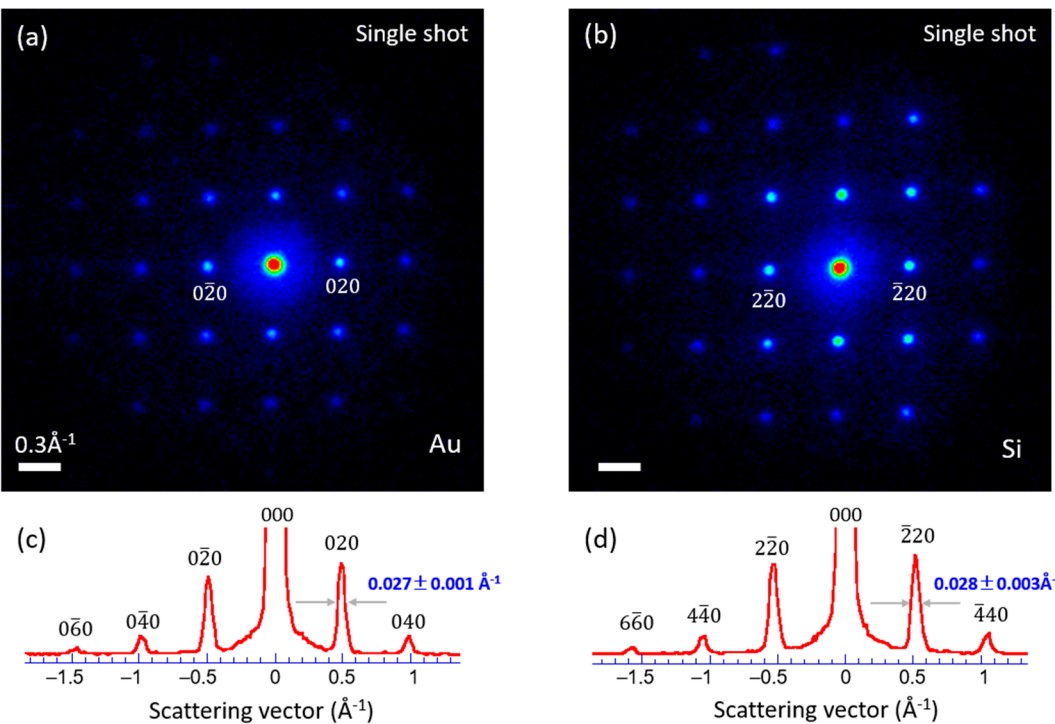

**Figure 2.** Single shot electron diffraction images of (**a**) (100)-orientated single-crystalline gold with a face-centered cubic structure and (**b**) (001)-orientated single-crystalline Si with a diamond structure, (**c**,**d**) are the intensity profiles of Bragg peaks along the $0\bar{2}0$ and 020 diffraction spots in the gold sample and along the $2\bar{2}0$ and $\bar{2}20$ diffraction spots in the Si sample, respectively. The number of electrons per pulse was $6.3 \times 10^4$. The electron energy was 2.5 MeV. The pulse duration was 55 ± 5 fs.

The data in Figure 2 show that (1) sharp diffraction patterns and good contrast can be obtained in the single shot measurement, (2) higher-order Bragg peaks of $0\bar{6}0$ in the single-crystalline gold and $6\bar{6}0$ in the single-crystalline Si with scattering vectors up to 1.5 Å$^{-1}$ can be detected, and (3) the width of the diffraction spots was measured to be 0.027 ± 0.001 Å$^{-1}$ for the single-crystalline gold and 0.028 ± 0.003 Å$^{-1}$ for the single-crystalline Si, indicating high spatial resolution. The obtained spatial resolution was determined mainly by the electron beam quality, e.g., convergence angle and energy spread. The relative energy spread was approximately $10^{-4}$ [23]. From the width of the diffraction spots and the distance of the diffraction spots from the 000 spot [28], the illumination convergence angle at the sample and the spatial coherence length of the electron beam were estimated to 55 ± 2 µrad and 5.6 ± 0.4 nm, respectively. This convergence angle is two orders of magnitude smaller than that of nonrelativistic UEDs. The spatial coherence length is also larger than that of nonrelativistic UEDs [29].

These enable one to observe sharp diffraction patterns and good-contrast diffraction images with a single shot, as shown in Figure 2.

### 3.2. Electron Diffraction of Polycrystalline Materials

In the electron diffraction imaging of polycrystalline materials, a polycrystalline aluminum foil with a thickness of 30 nm (EM-Japan) and a sample of thallous chloride chemical compound (EM-Japan) were used. The polycrystals contribute diffraction rings in the scintillator, resulting in the requirement of many electrons in the diffraction image. For the electron diffraction imaging of polycrystalline materials, the electron pulses were collimated by a large condenser aperture with a diameter of 1 mm. The number of electrons per pulse was $6.3 \times 10^5$ (100 fC) on the samples. Figure 3 shows the electron diffraction images of the polycrystalline aluminum (top images) and the thallous chloride crystals (bottom images) observed with single shot, 10-, and 100-pulse integrations. The electron energy was 2.5 MeV.

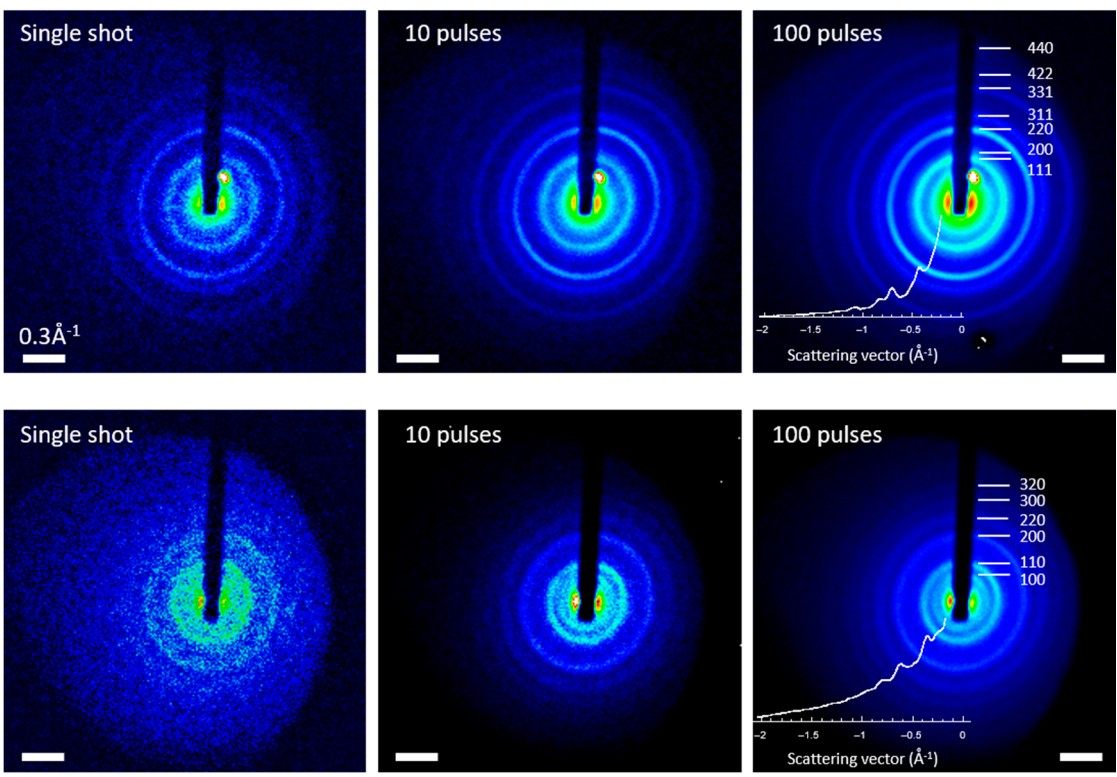

**Figure 3.** Electron diffraction images of the polycrystalline aluminum (top images) and the thallous chloride crystals (bottom images) observed with single shot, 10-, and 100-pulse integrations. The lines in the right hand images represent the intensity profiles of Bragg peaks of the polycrystalline aluminum and the thallous chloride polycrystals. The number of electrons per pulse was $6.3 \times 10^5$. The electron energy was 2.5 MeV.

The data in Figure 3 indicate that our compact relativistic UED enables the electron diffraction imaging of polycrystalline materials. The high-quality diffraction images were obtained with 10 pulses, and the entire diffraction images were detected with 100-pulse integration. Moreover, the single shot electron diffraction was available for the polycrystalline aluminum and the thallous chloride crystals.

### 3.3. Electron Diffraction of Amorphous Materials

In the electron diffraction imaging of amorphous materials, we used an amorphous germanium sample. The amorphous germanium with a thickness of 20 nm was evaporated on a 20-nm-thick silicon nitride membrane patterned with 2-μm-diameter pores (Cat. No. GE20-SN20MP2Q05, Alliance

Biosystems). For the electron diffraction imaging of the amorphous germanium, the electron pulses were collimated by a condenser aperture with a diameter of 1 mm. The number of electrons per pulse was $6.3 \times 10^5$, as in the observation of polycrystals. Figure 4 shows the electron diffraction images of the amorphous germanium obtained with single shot, 10-, and 100-pulse integrations. The electron energy was 2.5 MeV. The diffuse ring patterns indicate that the sample of germanium is amorphous.

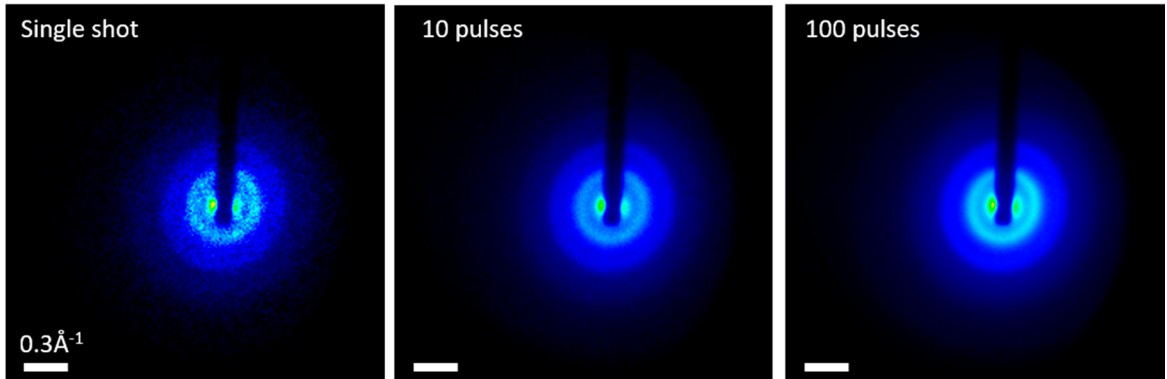

**Figure 4.** Electron diffraction images of amorphous germanium observed with single shot, 10-, and 100-pulse integrations. The number of electrons per pulse was $6.3 \times 10^5$. The electron energy was 2.5 MeV.

The experimental results indicate that our compact relativistic UED enables the electron diffraction imaging of amorphous state. In the measurement of amorphous germanium, high-quality diffraction image was obtained with 10 pulses, and the entire diffraction image was detected with 100-pulse integration. The single shot electron diffraction of amorphous state is available in our compact relativistic UED. The success on the single shot diffraction from polycrystalline and amorphous materials paves the way for studying irreversible phase transitions between the crystalline and amorphous states.

### 3.4. Time-Resolved Measurement of Single-Crystal Silicon

The intensities of Bragg peaks in the diffraction images reflect the lattice temperature effect and the structural information. By detecting the time evolution of the intensities of Bragg peaks, we can investigate structural dynamics in materials, e.g., photo-induced phase transitions [22].

In the time-resolved UED measurement, we used the (001)-orientated single crystalline Si sample described above. The thickness of Si sample was 35 nm. The single crystalline Si was excited by the pump laser pulses with the wavelength of 400 nm and the pulse duration of 54 fs RMS. The incident fluence of the pump laser was 14 mJ/cm$^2$. The spot size on the sample was 0.8 mm. The electron pulses were collimated by a condenser aperture with a diameter of 0.3 mm. The probe beam size of 0.3 mm on the sample is smaller than the size of the pump laser. The number of electrons per pulse was $6.3 \times 10^4$. We observed single shot electron diffraction images by changing the time delay of the pump laser pulse, and then detected the time evolution of the intensities of diffraction spots. Figure 5 shows the time evolution of the intensities of 220 and $\bar{2}20$ diffraction spots. The time step in the measurement was 66.6 fs adjusting the laser optical delay in the pump laser line. The characteristics of the probe electron pulses are same in the single shot electron diffraction measurement as described as Section 3.1. However, in the time-resolved measurement, the fluctuation of the electron number per pulse was approximately 0.9%, which was caused mainly by the fluctuation of the incident laser on the photocathode in the rf gun. The error bars of the data in Figure 5 represent the fluctuation of the electron number. The oscillation of the intensity at >1.5 ps was also due to the incident laser. The data are in agreement with an exponential decay with a time constant of 600 ± 50 fs, as shown as the line in Figure 5. Compared with the thermal relaxation mechanisms in crystalline silicon that transfer heat

from the hot carriers to the lattice on the time scale of a few picoseconds [30], the obtained ultrafast dynamics with the 600 ± 50 fs decay time constant are caused mainly by electronically driven phase transition in crystalline silicon, which agreed with the previous study [31].

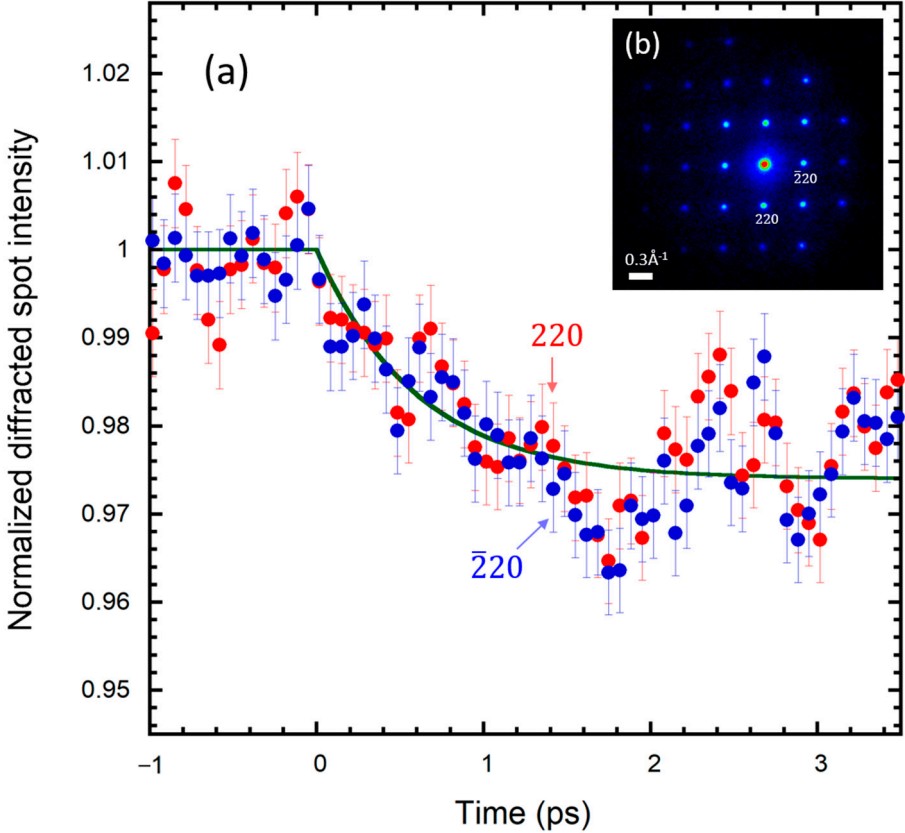

**Figure 5.** (**a**) The normalized intensities of the 220 and $\overline{2}20$ diffraction spots as a function of the time delay under the incident laser fluence of 14 mJ/cm$^2$, and (**b**) the single shot diffraction image of single crystalline Si before the laser excitation as same as in Figure 2b. The line represents a function of single exponential decay with a time constant of 600 ± 50 fs.

In the relativistic UED, the temporal resolution can be estimated by

$$\delta t = \sqrt{\sigma_e^2 + \sigma_l^2 + \Delta t_j^2}, \tag{1}$$

where $\sigma_e$ is the RMS duration of the electron pulse, $\sigma_l$ is the RMS duration of the pump laser pulse, and $\Delta t_j$ is the RMS time jitter between two pulses. In our compact relativistic UED, $\sigma_e = 55$ fs, $\sigma_l = 54$ fs, and $\Delta t_j = 61$ fs. Therefore, we estimated the RMS temporal resolution to $\delta t \approx 100$ fs, which is satisfactory to observe the ultrafast diffracted Bragg peak intensity decays with the time constant of 600 ± 50 fs in the single crystalline silicon sample.

## 4. Conclusions

In summary, we have developed a compact relativistic femtosecond electron diffractometer using an rf gun and an electron lens system. The rf gun generated femtosecond electron pulses with a duration of 55 ± 5 fs containing $10^4$~$10^5$ electrons per pulse at 2.5 MeV. The electron lens system provided good-contrast electron diffraction patterns with single shot, yielding a high spatial resolution of 0.027 ± 0.001 Å$^{-1}$. In the time-resolved UED imaging, we succeeded to observe the ultrafast electronically driven phase transition in single crystalline Si. The temporal resolution was approximately 100 fs.

In the single shot electron diffraction imaging of single crystalline silicon and gold, the sharp diffraction patterns and good contrast images were observed with a relativistic femtosecond electron pulse. The higher-order Bragg reflections with scattering vectors up to 1.5 Å$^{-1}$ were detected. In the electron diffraction imaging of polycrystalline and amorphous materials, high-quality diffraction images were obtained with 10 pulses, and the entire diffraction images were clearly detected with 100-pulse integration. Moreover, the single shot electron diffraction is also available for observing the polycrystalline and amorphous states.

The success in the UED imaging of the single-crystalline, polycrystalline, and amorphous states with relativistic electron pulses paves the way for studying photoinduced insulate-to-metal or semiconductor-to-metal phase transitions in metal oxide, i.e., vanadium dioxide and titanium oxide, and crystalline-to-amorphous phase transitions in chalcogenide materials, i.e., well-known $Ge_2Sb_2Te_5$. The single shot electron diffraction with relativistic femtosecond electron pulse is promising for studying irreversible processes, i.e., laser-excited melting dynamics and photoinduced chemical reactions in molecules on femtosecond/picosecond timescale.

However, in the relativistic UED demonstrations, the minimum electron beam size on the sample was 300 μm with a 300-μm-diameter condenser aperture, which is still unsuitable for microcrystal electron diffraction (micro ED) or now emerging three-dimensional electron diffraction (3D ED) [32]. In the next challenge, we propose a small condenser aperture (50 μm or less) to collimate the electron beam and then focus it on the sample with nanometer size by the condenser lens. The relativistic diffractometer with nanobeams will be useful for micro ED and 3D ED because of the long extinction distance of high-energy electrons.

**Author Contributions:** Project administration, J.Y.; preparation of samples, S.S.; Measurement of electron diffraction images, K.G.; time-resolved measurement, N.N.; discussions, Y.Y. All authors have read and agreed to the published version of the manuscript.

**Funding:** This research was funded by JSPS KAKENHI Grant Numbers JP22246127, JP26246026, and JP17H01060 of Grant-in-Aid for Scientific Research (A), and JP16K13687 of Challenging Research Exploratory, Japan at https://researchmap.jp/read0134206.

**Acknowledgments:** The authors acknowledge Kan K. and Gohdo M. of the Institute of Scientific and Industrial Research in Osaka University for their valuable discussions, Tanimura K. and Yasuda H. of the Research Center for ultra-high-voltage electron microscopy (UHVEM) in Osaka University for their valuable suggestions. Additionally, the authors thank Urakawa J., Takatomi T., and Terunuma N. of the High Energy Accelerator Research Organization (KEK) for the fabrication of the rf gun.

**Conflicts of Interest:** The authors declare no conflict of interest. The funders had no role in the design of the study; in the collection, analyses, or interpretation of data; in the writing of the manuscript; or in the decision to publish the results.

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
