# Peer review of "A Compact Ultrafast Electron Diffractometer with Relativistic Femtosecond Electron Pulses"

_qubs, doi:10.3390/qubs4010004_

Round 1

Reviewer 1 Report

The authors have described a compact UED diffractometer and have shown the possibilities of application of such diffractometer in uni-crystalline, poly-crystalline and amorphous materials. Possibilities of time-resolved studies of  ultrafast process has been discussed. The article can be accepted with minor revisions.

1. Reading the introductory text, the differences between the existing relativistic UED instrument (references 13-15) and the instrument developed by the authors is not immediately understandable. The phrase "electron lens system"  in the last paragraph of the introduction needs to be elaborated.

2. The number of electrons mentioned in the Fig caption 3 and 4 and the corresponding number of electrons mentioned in the text mismatches. The authors should mention the correct one.

3. The relativistic UED, as described by the authors, is useful for the study of irreversible ultrafast processes. Quantification of the structural information content in the single-shot diffraction from polycrystalline and amorphous material would be a valuable information.

4. There are many typos in the text such as fluence is written as "fluency", electron as "elecreon". Authors need to carefully go through the text and correct them.

Author Response

Dear Reviewer,

Thank you very much for your kind review.

We made a revision of our manuscript according to your kind comments and suggestions.

Yours sincerely,

Jinfeng Yang

-------------------------------------------------------------------------------------------------------------------------

Reading the introductory text, the differences between the existing relativistic UED instrument (references 13-15) and the instrument developed by the authors is not immediately understandable. The phrase "electron lens system" in the last paragraph of the introduction needs to be elaborated.

> We explained the electron lens system, and rewritten the paragraph in lines 47-56, page 2.

The number of electrons mentioned in the Fig caption 3 and 4 and the corresponding number of electrons mentioned in the text mismatches. The authors should mention the correct one.

> We corrected the number of electrons in the figure caption 3 and 4.

The relativistic UED, as described by the authors, is useful for the study of irreversible ultrafast processes. Quantification of the structural information content in the single-shot diffraction from polycrystalline and amorphous material would be a valuable information.

> Yes. We emphasized the single-shot diffraction from polycrystalline and amorphous materials and rewritten the paragraph in lines 182-188, page 5.

There are many typos in the text such as fluence is written as "fluency", electron as "elecreon". Authors need to carefully go through the text and correct them.

> We corrected them.

Reviewer 2 Report

Title: A compact ultrafast electron diffractometer with relativistic femtosecond electron pulses

Authors: Jinfeng Yang, Kazuki Gen, Nobuyasu Naruse, Shouichi Sakakihara, and Yoichi Yoshida

Synopsis: the paper describes implementation and first test applications of a new compact relativistic femtosecond electron pulse diffractometer.

General comment

The paper is well written and clear. The instrument is well-described and different kinds of results are shown. Still, authors should spend a bit more words for describing their results and compare them with similar techniques, like conventional and 3D electron diffraction.

Major comments

Authors should clearly state if the described instrument is the one already reported in Yang & Yoshida 2019 (reference 19). In that paper, diffraction tests were reported much more briefly, so this will not diminish the value of the current paper. Page 3, line 123 and Figure 2: reflections <200> should be extinct in silicon (space group Fd-3m). They normally appear in electron diffraction due to dynamical effects. In Figure 2, it is evident that reflections 0-20 and 020 appear and are very strong. Moreover, all reflection intensities look similar, with a linear reduction from the center to the peripheral ranges of the diffraction pattern. Looking at these patterns, I would say that dynamical effects are extremely strong and all information about structure factor is lost. So, these data can be used for cell parameter determination, but not for atomic structure determination or refinement. Can authors comment on this, considering also that they claim to have a very long extinction distance thanks to the high energy? Pages 4-5, chapter 3.2: which (crystallographic) resolution can author attain? Is the resolution an instrumental or sample limit? In conventional electron diffraction experiments by TEM on inorganic materials, resolutions of 0.7 Å or better are normal expected. I understand there is no possibility of sample tilt in this instrument. This make it very different from the now emerging 3D electron diffraction / MicroED technique (for a general review you can see Gemmi et al. 2019, ACS Cent Sci 5, 1315). Can authors comment on this, underling advantages/disadvantages of their method? Page 5, chapter 5: recently some authors proposed electron diffraction for PDF study (see for example Gorelik et al. 2019, Acta Crystallogr. B 75, 532). Can author compare their results to the ones reported in these papers. Page 7, Figure 5: the regression looks fine up to 1.4 ps. Then experimental points get quite random. Can authors comment on this? Moreover, it would be good to have a quantitative estimation of error for the decay value reported in Chapter 3.4. Page 7, Conclusions: authors are very generic when they speak about possible applications of their instrument/method: ‘The success in the UED imaging of the single-crystalline, polycrystalline and amorphous states with relativistic electron pulses paves the way for studying various ultrafast phase transformations, i.e. orthorhombic-monoclinic crystalline phase transitions and crystalline-amorphous phase transitions. The single shot electron diffraction with relativistic femtosecond electron pulse is promising for studying irreversible processes, i.e., irreversible phase transitions, chemical reactions, and dynamics of biomolecules on femtosecond timescale.’

Can authors list a series of concrete applications where their instrument/method is really important?

Minor comments

Page 1, line 38: ‘microscope’ probably better than

Author Response

Dear Reviewer,

Thank you very much for your kind review.

We made a revision of our manuscript according to your kind comments and suggestions.

Yours sincerely,

Jinfeng Yang

-----------------------------------------------------------------------------------------------------------------------------

General comment

The paper is well written and clear. The instrument is well-described and different kinds of results are shown. Still, authors should spend a bit more words for describing their results and compare them with similar techniques, like conventional and 3D electron diffraction.

We have made a revision of our manuscript according your kind comment.

Major comments

Authors should clearly state if the described instrument is the one already reported in Yang & Yoshida 2019 (reference 19). In that paper, diffraction tests were reported much more briefly, so this will not diminish the value of the current paper.

We added the following sentence in lines 103-106, page 3.

“Although the relativistic-energy electron diffraction imaging can be realized in the relativistic UEM [19], the presented relativistic UED instrument exhibits many excellent advantages, including ultrahigh vacuum at 10-9 Pa, sample preparation, and a large sample room for various in-situ observations with heating, pressuring, electric field, liquid or gas flow.”

Page 3, line 123 and Figure 2: reflections <200> should be extinct in silicon (space group Fd-3m). They normally appear in electron diffraction due to dynamical effects. In Figure 2, it is evident that reflections 0-20 and 020 appear and are very strong. Moreover, all reflection intensities look similar, with a linear reduction from the center to the peripheral ranges of the diffraction pattern. Looking at these patterns, I would say that dynamical effects are extremely strong and all information about structure factor is lost. So, these data can be used for cell parameter determination, but not for atomic structure determination or refinement. Can authors comment on this, considering also that they claim to have a very long extinction distance thanks to the high energy?

The reflections <200> are only shown in single-crystalline gold sample only in line 127 and Figs. 2(a) and (c), page 4. The dynamical effect was included in the diffraction pattern. However, the thickness of both samples is much less than the extinction distance of high-energy electrons. Our previous photo-excited gold melting study [21, 22] indicates that the kinematical diffraction description with the assumption of single elastic scattering events can be applied in the relativistic UED experiment.

The reduction from the center to the peripheral ranges of the diffraction pattern is caused mainly by the Gaussian distribution of the electrons with the width of 300 mm.

In the next experiments, we will measure the diffraction patterns with changing the silicon thickness. CBED will be also try to study in the next experiments.

Pages 4-5, chapter 3.2: which (crystallographic) resolution can author attain? Is the resolution an instrumental or sample limit? In conventional electron diffraction experiments by TEM on inorganic materials, resolutions of 0.7 Å or better are normal expected. I understand there is no possibility of sample tilt in this instrument. This make it very different from the now emerging 3D electron diffraction / MicroED technique (for a general review you can see Gemmi et al. 2019, ACS Cent Sci 5, 1315). Can authors comment on this, underling advantages/disadvantages of their method?

The resolution is an instrumental limit. We added the sentence of “The obtained spatial resolution was determined mainly by the electron beam quality, e.g. convergence angle and energy spread. The relative energy spread was approximately 10-4 [23].” in lines 133-135, page 4. The relativistic diffractometer is still unsuitable for micro ED or 3D ED because the electron beam size is rather large. In the next challenge, we propose a small condenser aperture to collimate the electron beam and then focus it on the sample with nanometer size. The relativistic diffractometer with nanobeams would be useful for micro ED and 3D ED because of the long extinction distance. We added the following paragraph in chapter 4, in lines 254-260, page 8.

“However, in the relativistic UED demonstrations, the minimum electron beam size on the sample was 300 m with a 300-m-diameter condenser aperture, which is still unsuitable for microcrystal electron diffraction (micro ED) or now emerging three-dimensional electron diffraction (3D ED) [32]. In the next challenge, we propose a small condenser aperture (50 m or less) to collimate the electron beam and then focus it on the sample with nanometer size by the condenser lens. The relativistic diffractometer with nanobeams will be useful for micro ED and 3D ED because of the long extinction distance of high-energy electrons.”

We also added a reference “Gemmi et al. 2019, ACS Cent Sci 5, 1315”.

Page 5, chapter 5: recently some authors proposed electron diffraction for PDF study (see for example Gorelik et al. 2019, Acta Crystallogr. B 75, 532). Can author compare their results to the ones reported in these papers.

The data of polycrystalline aluminum and amorphous germanium may be possible to be analyzed with ePDF. In this time, we cannot give a compassion of the results with the reported PDF study.

Page 7, Figure 5: the regression looks fine up to 1.4 ps. Then experimental points get quite random. Can authors comment on this? Moreover, it would be good to have a quantitative estimation of error for the decay value reported in Chapter 3.4.

The decay was fine up to 1.4~1.6 ps. We fit the decay with a single exponential function and obtained a time constant of the decay to 600 ± 50 fs. The error is obtained by fitting the experimental data. In this measurement, the fluctuation of electron number per pulse was about 0.9%, which is mainly caused by the fluctuation of incident laser on the photocathode in rf gun. The maximum change of the intensities of Bragg peaks is 2.5% in Figure 5, resulting in the random experimental data.

We added the error bars in Figure5 and the following sentence in lines 209-213, page 6.

“However, in the time-resolved measurement, the fluctuation of the electron number per pulse was approximately 0.9%, which was caused mainly by the fluctuation of the incident laser on the photocathode in the rf gun. The error bars of the data in Fig. 5 represent the fluctuation of the electron number. The oscillation of the intensity at >1.5 ps was also due to the incident laser.”

Page 7, Conclusions: authors are very generic when they speak about possible applications of their instrument/method: ‘The success in the UED imaging of the single-crystalline, polycrystalline and amorphous states with relativistic electron pulses paves the way for studying various ultrafast phase transformations, i.e. orthorhombic-monoclinic crystalline phase transitions and crystalline-amorphous phase transitions. The single shot electron diffraction with relativistic femtosecond electron pulse is promising for studying irreversible processes, i.e., irreversible phase transitions, chemical reactions, and dynamics of biomolecules on femtosecond timescale.’ Can authors list a series of concrete applications where their instrument/method is really important?

We rewritten the conclusions: “The success in the UED imaging of the single-crystalline, polycrystalline and amorphous states with relativistic electron pulses paves the way for studying photoinduced insulate-to-metal or semiconductor-to-metal phase transitions in metal oxide, i.e. vanadium dioxide and titanium oxide, and crystalline-to-amorphous phase transitions in chalcogenide materials, i.e. well-known Ge2Sb2Te5. The single shot electron diffraction with relativistic femtosecond electron pulse is promising for studying irreversible processes, i.e., laser-excited melting dynamics and photoinduced chemical reactions in molecules on femtosecond/picosecond timescale.”

Minor comments

Page 1, line 38: ‘microscope’ probably better than

We used ‘microscope’ in line 38.

Round 2

Reviewer 2 Report

Authors adequately replied all scientific issues. The paper is now ready for publication.

I have only a small curiosity: at page 3 authors added that their UED can be used for sample preparation. Can they explain better this claim?